# A Projection-free Algorithm for Constrained Stochastic Multi-level Composition Optimization

**Tesi Xiao**
Department of Statistics
University of California, Davis
texiao@ucdavis.edu

**Krishnakumar Balasubramanian**
Department of Statistics
University of California, Davis
kbala@ucdavis.edu

**Saeed Ghadimi**
Department of Management Sciences
University of Waterloo
sghadimi@uwaterloo.ca

## Abstract

We propose a projection-free conditional gradient-type algorithm for smooth stochastic multi-level composition optimization, where the objective function is a nested composition of $T$ functions and the constraint set is a closed convex set. Our algorithm assumes access to noisy evaluations of the functions and their gradients, through a stochastic first-order oracle satisfying certain standard unbiasedness and second-moment assumptions. We show that the number of calls to the stochastic first-order oracle and the linear-minimization oracle required by the proposed algorithm, to obtain an $\epsilon$-stationary solution, are of order $\mathcal{O}_T(\epsilon^{-2})$ and $\mathcal{O}_T(\epsilon^{-3})$ respectively, where $\mathcal{O}_T$ hides constants in $T$. Notably, the dependence of these complexity bounds on $\epsilon$ and $T$ are separate in the sense that changing one does not impact the dependence of the bounds on the other. For the case of $T = 1$, we also provide a high-probability convergence result that depends poly-logarithmically on the inverse confidence level. Moreover, our algorithm is parameter-free and does not require any (increasing) order of mini-batches to converge unlike the common practice in the analysis of stochastic conditional gradient-type algorithms.

## 1 Introduction

We study projection-free algorithms for solving the following stochastic multi-level composition optimization problem

$$\min_{x \in \mathcal{X}} \quad F(x) := f_1 \circ \cdots \circ f_T(x), \tag{1}$$

where $f_i : \mathbb{R}^{d_i} \to \mathbb{R}^{d_{i-1}}, i = 1, \cdots, T \ (d_0 = 1)$ are continuously differentiable functions, the composite function $F$ is bounded below by $F^\star > -\infty$ and $\mathcal{X} \subset \mathbb{R}^d$ is a closed convex set. We assume that the exact function values and derivatives of $f_i$'s are not available. In particular, we assume that $f_i(y) = \mathbb{E}_{\xi_i}[G_i(y, \xi_i)]$ for some random variable $\xi_i$. Our goal is to solve the above optimization problem, given access to noisy evaluations of $\nabla f_i$'s and $f_i$'s.

There are two main challenges to address in developing efficient projection-free algorithms for solving (1). First, note that denoting the transpose of the Jacobian of $f_i$ by $\nabla f_i$, the gradient of the objective function $F(x)$ in (1), is given by $\nabla F(x) = \nabla f_T(y_T) \nabla f_{T-1}(y_{T-1}) \cdots \nabla f_1(y_1)$, where $y_i = f_{i+1} \circ \cdots \circ f_T(x)$ for $1 \leq i < T$, and $y_T = x$. Because of the nested nature of the gradient $\nabla F(x)$, obtaining an unbiased gradient estimator in the stochastic first-order setting, with controlled moments, becomes non-trivial. Using naive stochastic gradient estimators lead to oracle complexities

36th Conference on Neural Information Processing Systems (NeurIPS 2022).

that depend exponentially on $T$ (in terms of the accuracy parameter). Next, even when $T = 1$ in the stochastic setting, projection-free algorithms like the conditional gradient method or its sliding variants invariably require increasing order of mini-batches[1] [28, 40, 24, 39, 52], which make their practical implementation infeasible.

In this work, we propose a projection-free conditional gradient-type algorithm that achieves *level-independent* oracle complexities (i.e., the dependence of the complexities on the target accuracy is independent of $T$) using only *one sample* of $(\xi_i)_{1 \le i \le T}$ in each iteration, thereby addressing both of the above challenges. Our algorithm uses moving-average based stochastic estimators of the gradient and function values, also used recently by [19] and [4], along with an inexact conditional gradient method used by [3] (which in turn is inspired by the sliding method by [28]). In order to establish our oracle complexity results, we use a novel merit function based convergence analysis. To the best of our knowledge, such an analysis technique is used for the first time in the context of analyzing stochastic conditional gradient-type algorithms.

## 1.1 Preliminaries and Main Contributions

We now introduce the technical preliminaries required to state and highlight the main contributions of this work. Throughout this work, $\|\cdot\|$ denotes the Euclidean norm for vectors and the Frobenius norm for matrices. We first describe the set of assumptions on the objective functions and the constraint set.

**Assumption 1** (Constraint set). *The set $\mathcal{X} \subset \mathbb{R}^d$ is convex and closed with $\max_{x,y \in \mathcal{X}} \|x - y\| \le D_{\mathcal{X}}$.*

**Assumption 2** (Smoothness). *All functions $f_1, \ldots, f_T$ and their derivatives are Lipschitz continuous with Lipschitz constants $L_{f_i}$ and $L_{\nabla f_i}$, respectively.*

The above assumptions on the constraint set and the objective function are standard in the literature on stochastic optimization, and in particular in the analysis of conditional gradient algorithms and multi-level optimization; see, for example, [28], [50] and [4]. We emphasize here that the above smoothness assumptions are made only on the functions $(f_i)_{1 \le i \le T}$ and not on the stochastic functions $(G_i)_{1 \le i \le T}$ (which would be a stronger assumption). Moreover, the Lipschitz continuity of $f_i$'s are implied by the Assumption 1 and assuming the functions are continuously differentiable. However, for sake of illustration, we state both assumptions separately. In addition to these assumptions, we also make unbiasedness and bounded-variance assumptions on the stochastic first-order oracle. Due to its technical nature, we state the precise details later in Section 3 (see Assumption 3).

We next turn our attention to the convergence criterion that we use in this work to evaluate the performance of the proposed algorithm. Gradient-based algorithms iteratively solve sub-problems in the form of

$$\arg\min_{u \in \mathcal{X}} \left\{ \langle g, u \rangle + \frac{\beta}{2} \|u - x\|^2 \right\}, \tag{2}$$

for some $\beta > 0$, $g \in \mathbb{R}^d$ and $x \in \mathcal{X}$. Denoting the optimal solution of the above problem by $P_{\mathcal{X}}(x, g, \beta)$ and noting its optimality condition, one can easily show that

$$-\nabla F(\bar{x}) \in \mathcal{N}_{\mathcal{X}}(\bar{x}) + \mathcal{B}\Big(0, \|g - \nabla F(\bar{x})\| D_{\mathcal{X}} + \beta \|x - P_{\mathcal{X}}(x, g, \beta)\|(D_{\mathcal{X}} + \|\nabla F(\bar{x})\|/\beta)\Big),$$

where $\mathcal{N}_{\mathcal{X}}(\bar{x})$ is the normal cone to $\mathcal{X}$ at $\bar{x}$ and $\mathcal{B}(0, r)$ denotes a ball centered at the origin with radius $r$. Thus, reducing the radius of the ball in the above relation will result in finding an approximate first-order stationary point of the problem for non-convex constrained minimization problems. Motivated by this fact, we can define the gradient mapping at point $\bar{x} \in \mathcal{X}$ as

$$\mathcal{G}_{\mathcal{X}}(\bar{x}, \nabla F(\bar{x}), \beta) := \beta\left(\bar{x} - P_{\mathcal{X}}(\bar{x}, \nabla F(\bar{x}), \beta)\right) = \beta\left(\bar{x} - \Pi_{\mathcal{X}}\left(\bar{x} - \frac{1}{\beta}\nabla F(\bar{x})\right)\right), \tag{3}$$

where $\Pi_{\mathcal{X}}(y)$ denotes the Euclidean projection of the vector $y$ onto the set $\mathcal{X}$. The gradient mapping is a classical measure has been widely used in the literature as a convergence criterion when solving nonconvex constrained problems [35]. It plays an analogous role of the gradient for constrained optimization problems; in fact when the set $\mathcal{X} \equiv \mathbb{R}^d$ the gradient mapping just reduces to $\nabla F(\bar{x})$. It should be emphasized that while the gradient mapping cannot be computed in the stochastic setting, it

---

[1]We discuss in detail about some recent works that avoid requiring increasing mini-batches, albeit under stronger assumptions, in Section 1.2.

| Algorithm | Criterion | # of levels | Batch size | SFO | LMO |
|---|---|---|---|---|---|
| SPIFER-SFW [52] | FW-gap | 1 | $\mathcal{O}(\epsilon^{-1})$ | $\mathcal{O}(\epsilon^{-3})$ | $\mathcal{O}(\epsilon^{-2})$ |
| 1-SFW [54] | FW-gap | 1 | 1 | $\mathcal{O}(\epsilon^{-3})$ | $\mathcal{O}(\epsilon^{-3})$ |
| SCFW [1] | FW-gap | 2 | 1 | $\mathcal{O}(\epsilon^{-3})$ | $\mathcal{O}(\epsilon^{-3})$ |
| SCGS [39] | GM | 1 | $\mathcal{O}(\epsilon^{-1})$ | $\mathcal{O}(\epsilon^{-2})$ | $\mathcal{O}(\epsilon^{-2})$ |
| SGD+ICG [3] | GM | 1 | $\mathcal{O}(\epsilon^{-1})$ | $\mathcal{O}(\epsilon^{-2})$ | $\mathcal{O}(\epsilon^{-2})$ |
| LiNASA+ICG (Algorithm 1) | GM | $T$ | 1 | $\mathcal{O}_T(\epsilon^{-2})$ | $\mathcal{O}_T(\epsilon^{-3})$ |

Table 1: Complexity results for stochastic conditional gradient type algorithms to find an $\epsilon$-stationary solution in the nonconvex setting. FW-Gap and GM stands for Frank-Wolfe Gap (see (4)) and Gradient Mapping (see (3)) respectively. $\mathcal{O}_T$ hides constants in $T$. Existing one-sample based stochastic conditional gradient algorithms are either (i) not applicable to the case of general $T > 1$, or (ii) require strong assumptions [54], or (iii) are not truly online [1]; see Section 1.2 for detailed discussion. The results in [3] are actually presented for the zeroth-order setting; however the above stated first-order complexities follow immediately.

still serves as a measure of convergence. Our main goal in this work is to find an $\epsilon$-stationary solution to (1), in the sense described below.

**Definition 1.** *A point $\bar{x} \in \mathcal{X}$ generated by an algorithm for solving* (1) *is called an $\epsilon$-stationary point, if we have $\mathbb{E}[\|\mathcal{G}_{\mathcal{X}}(\bar{x}, \nabla F(\bar{x}), \beta)\|^2] \leq \epsilon$, where the expectation is taken over all the randomness involved in the problem.*

In the literature on conditional gradient methods for the nonconvex setting, the so-called Frank-Wolfe gap is also widely used to provide convergence analysis. The Frank-Wolfe Gap is defined formally as

$$g_{\mathcal{X}}(\bar{x}, \nabla F(\bar{x})) := \max_{y \in \mathcal{X}} \langle \nabla F(\bar{x}), \bar{x} - y \rangle. \tag{4}$$

As pointed out by [3], the gradient mapping criterion and the Frank-Wolfe gap are related to each other in the following sense.

**Proposition 1.** [3] *Let $g_{\mathcal{X}}(\cdot)$ be the Frank-Wolfe gap defined in* (4) *and $\mathcal{G}_{\mathcal{X}}(\cdot)$ be the gradient mapping defined in* (3). *Then, we have $\|\mathcal{G}_{\mathcal{X}}(x, \nabla F(x), \beta)\|^2 \leq g_{\mathcal{X}}(x, \nabla F(x)), \forall x \in \mathcal{X}$. Moreover, under Assumption 1, 2, we have $g_{\mathcal{X}}(x, \nabla F(x)) \leq \left[(1/\beta) \prod_{i=1}^{T} L_{f_i} + D_{\mathcal{X}}\right] \|\mathcal{G}_{\mathcal{X}}(x, \nabla F(x), \beta)\|$.*

For stochastic conditional gradient-type algorithms, the oracle complexity is measured in terms of number of calls to the Stochastic First-order Oracle (SFO) and the Linear Minimization Oracle (LMO) used to the solve the sub-problems (that are of the form of minimizing a linear function over the convex feasible set) arising in the algorithm. In this work, we hence measure the number of calls to SFO and LMO required by the proposed algorithm to obtain an $\epsilon$-stationary solution in the sense of Definition 1. We now highlight our **main contributions**:

- We propose a projection-free conditional gradient-type method (Algorithm 1) for solving (1). In Theorem 2, we show that the SFO and LMO complexities of this algorithm, in order to achieve an $\epsilon$-stationary solution in the sense of Definition 1, are of order $\mathcal{O}(\epsilon^{-2})$ and $\mathcal{O}(\epsilon^{-3})$, respectively.

- The above SFO and LMO complexities are in particular level-independent (i.e., the dependence of the complexities on the target accuracy is independent of $T$). The proposed algorithm is parameter-free and does not require any mini-batches, making it applicable for the online setting.

- When considering the case of $T \leq 2$, we present a simplified method (Algorithm 3 and 4) to obtain the same oracle complexities. Intriguingly, while this simplified method is still parameter-free for $T = 1$, it is not when $T = 2$ (see Theorem 3 and Remark 3.1). Furthermore, for the case of $T = 1$, we also establish high-probability bounds (see Theorem 5).

A summary of oracle complexities for stochastic conditional gradient-type algorithms is in Table 1.

## 1.2 Related Work

**Conditional Gradient-Type Method.** The conditional gradient algorithm [15, 29], has had a renewed interest in the machine learning and optimization communities in the past decade; see [33,

26, 20, 27, 5, 17] for a partial list of recent works. Considering the stochastic convex setup, [22, 24] provided expected oracle complexity results for the stochastic conditional gradient algorithm. The complexities were further improved by a sliding procedure in [28], based on Nesterov's acceleration method. [40, 52, 24] considered variance reduced stochastic conditional gradient algorithms, and provided expected oracle complexities in the non-convex setting. [39] analyzed the sliding algorithm in the non-convex setting and provided results for the gradient mapping criterion. *All of the above works require increasing orders of mini-batches to obtain their oracle complexity results.*

[34] and [54] proposed a stochastic conditional gradient-type algorithm with moving-average gradient estimator for the convex and non-convex setting that uses only one-sample in each iteration. However, [34] and [54] require several restrictive assumptions, which we explain next (focusing on [54] which considers the nonconvex case). Specifically, [54] requires that the stochastic function $G_1(x, \xi_1)$ has uniformly bounded function value, gradient-norm, and Hessian spectral-norm, and the distribution of the random vector $\xi_1$ has an absolutely continuous density $p$ such that the norm of the gradient of $\log p$ and spectral norm of the Hessian of $\log p$ has finite fourth and second-moments respectively. In contrasts, we do not require such stringent assumptions. Next, all of the above works focus only on the case of $T = 1$. [1] considered stochastic conditional gradient algorithm for solving (1), with $T = 2$. However, [1] also makes stringent assumptions: (i) the stochastic objective functions $G_1(x, \xi_1)$ and $G_2(x, \xi_1)$ themselves have Lipschitz gradients almost surely and (ii) for a given instance of random vectors $\xi_1$ and $\xi_2$, one could query the oracle at the current and previous iterations, which makes the algorithm not to be truly online. See Table 1 for a summary.

**Stochastic Multi-level Composition Optimization.** Compositional optimization problems of the form in (1) have been considered as early as 1970s by [12]. Recently, there has been a renewed interest on this problem. [13] and [10] considered a sample-average approximation approach for solving (1) and established several asymptotic results. For the case of $T = 2$, [48], [49] and [6] proposed and analyzed stochastic gradient descent-type algorithms in the smooth setting. [9] and [11] considered the non-smooth setting and established oracle complexity results. Furthermore, [25] proposed algorithms when the randomness between the two levels are not necessarily independent. For the general case of $T \geq 1$, [50] proposed stochastic gradient descent-type algorithms and established oracle complexities established that depend exponentially on $T$ and are hence sub-optimal. Indeed, large deviation and Central Limit Theorem results established in [13, 10], respectively, show that in the sample-average approximation setting, the $\arg\min$ of the problem in (1) based on $n$ samples, converges at a level-independent rate (i.e., dependence of the convergence rate on the target accuracy is independent of $T$) to the true minimizer, under suitable regularity conditions.

[19] proposed a single time-scale Nested Averaged Stochastic Approximation (NASA) algorithm and established optimal rates for the cases of $T = 1, 2$. For the general case of $T \geq 1$, [4] proposed a linearized NASA algorithm and established level-independent and optimal convergence rates. Concurrently, [43] considered the case when the function $f_i$ are non-smooth and established asymptotic convergence results. [53] also established non-asymptotic level-independent oracle complexities, however, under stronger assumptions than that in [4]. Firstly, they assumed that for a fixed batch of samples, one could query the oracle on different points, which is not suited for the general online stochastic optimization setup. Next, they assume a much stronger mean-square Lipschitz smoothness assumption on the individual functions $f_i$ and their gradients. Finally, they required mini-batches sizes that depend exponentially on $T$, which makes their method impractical. Concurrent to [4], level-independent rates were also obtained for *unconstrained* problems by [7], albeit, under the stronger assumption that the stochastic functions $G_i(x, \xi_i)$ are Lipschitz, almost surely. It is also worth mentioning that while some of the above papers considered constrained problems, the algorithms proposed and analyzed in the above works are not projection-free, which is the main focus of this work.

## 2    Methodology

In this section, we present our projection-free algorithm for solving problem (1). The method generates three random sequences, namely, approximate solutions $\{x^k\}$, average gradients $\{z^k\}$, and average function values $\{u^k\}$, defined on a certain probability space $(\Omega, \mathscr{F}, P)$. We let $\mathscr{F}_k$ to be the $\sigma$-algebra generated by $\{x^0, \ldots, x^k, z^0, \ldots, z^k, u_1^0, \ldots, u_1^k, \ldots, u_T^0, \ldots, u_T^k\}$. The overall method is given in Algorithm 1. In (7), the stochastic Jacobians $J_i^{k+1} \in \mathbb{R}^{d_i \times d_{i-1}}$, and the product

**Algorithm 1** Linearized NASA with Inexact Conditional Gradient Method (`LiNASA+ICG`)
___
**Input:** $x^0 \in \mathcal{X}$, $z^0 = 0 \in \mathbb{R}^d$, $u_i^0 \in \mathbb{R}^{d_i}$, $i = 1, \dots, T$, $\beta_k > 0$, $t_k > 0$, $\tau_k \in (0, 1]$, $\delta \geq 0$.
**for** $k = 0, 1, 2, \dots, N$ **do**
    1. Update the solution:

$$\tilde{y}^k = \text{ICG}(x^k, z^k, \beta_k, t_k, \delta), \tag{5}$$

$$x^{k+1} = x^k + \tau_k(\tilde{y}^k - x^k), \tag{6}$$

    and compute stochastic Jacobians $J_i^{k+1}$, and function values $G_i^{k+1}$ at $u_{i+1}^k$ for $i = 1, \dots, T$.
    2. Update average gradients $z$ and function value estimates $u_i$ for each level $i = 1, \dots, T$

$$z^{k+1} = (1 - \tau_k)z^k + \tau_k \prod_{i=1}^{T} J_{T+1-i}^{k+1}, \tag{7}$$

$$u_i^{k+1} = (1 - \tau_k)u_i^k + \tau_k G_i^{k+1} + \langle J_i^{k+1}, u_{i+1}^{k+1} - u_{i+1}^k \rangle. \tag{8}$$

**end for**
**Output:** $(x^R, z^R, u_1^R, \cdots, u_T^R)$, where $R$ is uniformly distributed over $\{1, 2, \dots, N\}$
___

**Algorithm 2** Inexact Conditional Gradient Method (`ICG`)
___
**Input:** $(x, z, \beta, M, \delta)$
**Set** $w^0 = x$.
**for** $t = 0, 1, 2, \dots, M$ **do**
    1. Find $v^t \in \mathcal{X}$ with a quantity $\delta \geq 0$ such that

$$\langle z + \beta(w^t - x), v^t \rangle \leq \min_{v \in \mathcal{X}} \langle z + \beta(w^t - x), v \rangle + \frac{\beta D_{\mathcal{X}}^2 \delta}{t + 2}.$$

    2. Set $w^{t+1} = (1 - \mu_t)w^t + \mu_t v^t$ with $\mu_t = \min\left\{1, \frac{\langle \beta(x - w^t) - z, v^t - w^t \rangle}{\beta \|v^t - w^t\|^2}\right\}$.

**end for**
**Output:** $w^M$
___

$\prod_{i=1}^{T} J_{T+1-i}^{k+1}$ is calculated as $J_T^{k+1} J_{T-1}^{k+1} \cdots J_1^{k+1} \in \mathbb{R}^{d_T \times d_1} \equiv \mathbb{R}^{d_T \times 1}$. In (8), we use the notation $\langle \cdot, \cdot \rangle$ to represent both matrix-vector multiplication and vector-vector inner product. There are two aspects of the algorithm that we highlight specifically: (**i**) In addition to estimating the gradient of $F$, we also estimate a stochastic linear approximation of the inner functions $f_i$ by a moving-average technique. In the multi-level setting we consider, it helps us to avoid the accumulation of bias, when estimating the $f_i$ directly. Linearization techniques were used in the stochastic optimization since the work of [42]. A similar approach was used in [4] in the context of projected-based methods for solving (1). It is also worth mentioning that other linearization techniques have been used in [9] and [11] for estimating the stochastic inner function values for weakly convex two-level composition problems, and (**ii**) The `ICG` method given in Algorithm 2 is essentially applying *deterministic* conditional gradient method with the exact line search for solving the quadratic minimization subproblem in (2) with the estimated gradient $z_k$ in (7). It was also used in [3] as a sub-routine and is motivated by the sliding approach of [28].

## 3  Main Results

In this section, we present our main result on the oracle complexity of Algorithm 1. Before we proceed, we present our assumptions on the stochastic first-order oracle.

**Assumption 3** (Stochastic First-Order Oracle). *Denote* $u_{T+1}^k \equiv x^k$. *For each $k$, $u_{i+1}^k$ being the input, the stochastic oracle outputs $G_i^{k+1} \in \mathbb{R}^{d_i}$ and $J_i^{k+1}$ such that given $\mathscr{F}_k$ and for any $i \in \{1, \dots, T\}$*

*(a)* $\mathbb{E}[J_i^{k+1}|\mathscr{F}_k] = \nabla f_i(u_{i+1}^k)$, $\mathbb{E}[G_i^{k+1}|\mathscr{F}_k] = f_i(u_{i+1}^k)$,

*(b)* $\mathbb{E}[\|G_i^{k+1} - f_i(u_{i+1}^k)\|^2 | \mathscr{F}_k] \leq \sigma_{G_i}^2$, $\mathbb{E}[\|J_i^{k+1} - \nabla f_i(u_{i+1}^k)\|^2 | \mathscr{F}_k] \leq \sigma_{J_i}^2$,

*(c)* *The outputs of the stochastic oracle at level $i$, $G_i^{k+1}$ and $J_i^{k+1}$, are independent. The outputs of the stochastic oracle are independent between levels, i.e., $\{G_i^{k+1}\}_{i=1,\ldots,T}$ are independent and so are $\{J_i^{k+1}\}_{i=1,\ldots,T}$.*

Parts (a) and (b) in Assumption 3 are standard unbiasedness and bounded variance assumptions on the stochastic gradient, common in the literature. Part (c) is essential to establish the convergence results in the multi-level case. Similar assumptions have been made, for example, in [50] and [4]. We also emphasize that unlike some prior works (see e.g., [54]), Assumption 3 allows the case of endogenous uncertainty, and we do not require the distribution of the random variables $(\xi_i)_{1 \leq i \leq T}$ to be independent of the distribution of the decision variables $(u_i)_{1 \leq i \leq T}$.

**Remark.** *Under Assumption 2, and 3, we can immediately conclude that $\mathbb{E}[\|J_i^{k+1}\|^2 | \mathscr{F}_k] = \mathbb{E}[\|J_i^{k+1} - \nabla f_i(u_{i+1}^k)\|^2 | \mathscr{F}_k] + \|\nabla f_i(u_{i+1}^k)\|^2 \leq \sigma_{J_i}^2 + L_{f_i}^2 := \hat{\sigma}_{J_i}^2$. In the sequel, $\hat{\sigma}_{J_i}^2$ will be used to simplify the presentation.*

We start with the merit function used in this work and its connection to the gradient mapping criterion. Our proof leverages the following merit function:

$$W_{\alpha,\gamma}(x, z, u) = F(x) - F^\star - \eta(x, z) + \alpha\|\nabla F(x) - z\|^2 + \sum_{i=1}^{T} \gamma_i \|f_i(u_{i+1}) - u_i\|^2, \quad (9)$$

where $\alpha, \{\gamma_i\}_{1 \leq i \leq T}$ are positive constants and

$$\eta(x, z) = \min_{y \in \mathcal{X}} \left\{ H(y; x, z, \beta) := \langle z, y - x \rangle + \frac{\beta}{2}\|y - x\|^2 \right\}. \quad (10)$$

Compared to [4], we require the additional term $\|\nabla F(x) - z\|^2$, which turns out to be essential in our proof due to the ICG routine. The following proposition relates the merit function above to the gradient mapping.

**Proposition 2.** *Let $\mathcal{G}_{\mathcal{X}}(\cdot)$ be the gradient mapping defined in (3) and $\eta(\cdot, \cdot)$ be defined in (10). For any pair of $(x, z)$ and $\beta > 0$, we have $\|\mathcal{G}_{\mathcal{X}}(x, \nabla F(x), \beta)\|^2 \leq -4\beta\eta(x, z) + 2\|\nabla F(x) - z\|^2$.*

*Proof.* By expanding the square, and using the properties of projection operation, we have

$$\|\Pi_{\mathcal{X}}(x - \frac{1}{\beta}z) - x\|^2 + \|\Pi_{\mathcal{X}}(x - \frac{1}{\beta}z) - (x - \frac{1}{\beta}z)\|^2 \leq \|\bar{x} - (x - \frac{1}{\beta}z)\|^2 = \|\frac{1}{\beta}z\|^2.$$

Thus, we have $\eta(x, z) \leq -\frac{\beta}{2}\|\Pi_{\mathcal{X}}(x - \frac{1}{\beta}z) - x\|^2$. The proof is completed immediately by noting that $\|\mathcal{G}(x, \nabla F(x), \beta)\|^2 \leq 2\beta^2 \|\Pi_{\mathcal{X}}(x - \frac{1}{\beta}z) - x\|^2 + 2\|\nabla F(x) - z\|^2$. $\qquad\square$

We now present out main result on the oracle complexity of Algorithm1.

**Theorem 2.** *Under Assumption 1, 2, 3, let $\{x^k, z^k, \{u_i^k\}_{1 \leq i \leq T}\}_{k \geq 0}$ be the sequence generated by Algorithm 1 with $N \geq 1$ and*

$$\beta_k \equiv \beta > 0, \qquad \tau_0 = 1, \ t_0 = 0, \quad \tau_k = \frac{1}{\sqrt{N}}, \ t_k = \lceil \sqrt{k} \rceil, \quad \forall k \geq 1, \quad (11)$$

*where $\beta$ is an arbitrary positive constant. Provided that the merit function $W_{\alpha,\gamma}(x, z, u)$ is defined as (9) with*

$$\alpha = \frac{\beta}{20L_{\nabla F}^2}, \quad \gamma_1 = \frac{\beta}{2}, \quad \gamma_j = \left(2\alpha + \frac{1}{4\alpha L_{\nabla F}^2}\right)(T-1)C_j^2 + \frac{\beta}{2}, \quad 2 \leq j \leq T, \quad (12)$$

*we have,*

$$\mathbb{E}\left[\|\mathcal{G}_{\mathcal{X}}(x^R, \nabla F(x^R), \beta)\|^2\right] \leq \frac{2(\beta + \frac{20L_{\nabla F}^2}{\beta})\left[2W_{\alpha,\gamma}(x^0, z^0, u^0) + \mathcal{B}(\beta, \sigma^2, L, D_{\mathcal{X}}, T, \delta)\right]}{\sqrt{N}},$$

$$(13)$$

$$\mathbb{E}\left[\|f_i(u_{i+1}^R) - u_i^R\|^2\right] \leq \frac{2W_{\alpha,\gamma}(x^0, z^0, u^0) + \mathcal{B}(\beta, \sigma^2, L, D_\mathcal{X}, T, \delta)}{\beta\sqrt{N}}, \quad 1 \leq i \leq T. \tag{14}$$

where $u_{T+1} = x, \mathcal{B}(\beta, \sigma^2, L, D_\mathcal{X}, T, \delta) = 4\hat{\sigma}^2 + 32\beta D_\mathcal{X}^2(1+\delta)\left(\frac{3}{5} + \frac{5L_{\nabla F}^2}{\beta^2}\right)$, and $\hat{\sigma}^2$ is a constant depending on the parameters $(\beta, \sigma^2, L, D_\mathcal{X}, T)$ given in (42). The expectation is taken with respect to all random sequences generated by the method and an independent random integer number $R$ uniformly distributed over $\{1, \ldots, N\}$. That is to say, the number of calls to SFO and LMO to get an $\epsilon$-stationary point is upper bounded by $\mathcal{O}_T(\epsilon^{-2}), \mathcal{O}_T(\epsilon^{-3})$ respectively.

**Remark.** *The constant $\mathcal{B}(\beta, \sigma^2, L, D_\mathcal{X}, T, \delta)$ is $\mathcal{O}(T)$ given the definition of $\hat{\sigma}^2$ and the value of $\gamma_j$ in (12), which further implies that the total number of calls to SFO and LMO of Algorithm 1 for finding an $\epsilon$-stationary point of (1), are bounded by $\mathcal{O}(T^2\epsilon^{-2}) = \mathcal{O}_T(\epsilon^{-2})$ and $\mathcal{O}(T^3\epsilon^{-3}) = \mathcal{O}_T(\epsilon^{-3})$ respectively. Furthermore, it is worth noting that this complexity bound for Algorithm 1 is obtained without any dependence of the parameter $\beta_k$ on Lipschitz constants due to the choice of arbitrary positive constant $\beta$ in (11), and $\tau_k, t_k$ depend only on the number of iterations $N$ and $k$ respectively. This makes Algorithm 1 parameter-free and easy to implement.*

**Remark.** *As discussed in Section 2, the `ICG` routine given in Algorithm 2 is a deterministic method with the estimated gradient $z_k$ in (7). The number of iterations, $t_k$, required to run Algorithm 2 is given by $t_k = \lceil\sqrt{k}\rceil$. That is, we require more precise solutions for the `ICG` routine, only for later outer iterations. Furthermore, due to the deterministic nature of the `ICG` routine, further advances in the analysis of deterministic conditional gradient methods under additional assumptions on the constraint set $\mathcal{X}$ (see, for example, [16, 18]) could be leveraged to improve the overall LMO complexity.*

### 3.1 The special cases of $T = 1$ and $T = 2$

We now discuss several intriguing points regarding the choice of tuning parameter $\beta$, for the case of $T = 2$, and the more standard case of $T = 1$. Specifically, the linearization technique used in Algorithm 1 turns out to be not necessary for the case of $T = 2$ and $T = 1$ to obtain similar rates. However, without linearization, the choice of $\beta$ is dependent on the problem parameters for $T = 2$. Whereas it turns out to be independent of the problem parameters (similar to Algorithm 1 and Theorem 2 which holds for all $T \geq 1$) for $T = 1$. As the outer function value estimates (i.e., $u_1^{k+1}$ sequence) are not required for the convergence analysis, we remove them in Algorithms 3 and 4.

---

**Algorithm 3** NASA with Inexact Conditional Gradient Method (`NASA+ICG`) for $T = 2$
___
Replace Step 2 of Algorithm 1 with the following:
2'. Update the average gradient $z$ and the function value estimate $u_2$ respectively as:

$$z^{k+1} = (1 - \tau_k)z^k + \tau_k J_2^{k+1}J_1^{k+1} \quad \text{and} \quad u_2^{k+1} = (1 - \tau_k)u^k + \tau_k G_2^{k+1}$$

---

**Algorithm 4** ASA with Inexact Conditional Gradient Method (`ASA+ICG`) for $T = 1$
___
Replace Step 2 of Algorithm 1 with the following:
2''. Update the average gradient $z$ as: $z^{k+1} = (1 - \tau_k)z^k + \tau_k J_1^{k+1}$.

---

**Theorem 3.** *Let Assumptions 1, 2, 3 be satisfied by the optimization problem (1). Let $\mathcal{C}_1, \mathcal{C}_2$ and $\mathcal{C}_3$ be some constants depending on the parameters $(\beta, \sigma^2, L, D_\mathcal{X}, \delta)$, as defined in (54) and (62). Let $\tau_0 = 1, t_0 = 0, \tau_k = \frac{1}{\sqrt{N}}, t_k = \lceil\sqrt{k}\rceil, \forall k \geq 1$, where $N$ is the total number of iterations.*

*(a) Let $T = 2$, and let $\{x^k, z^k, u_2^k\}_{k\geq0}$ be the sequence generated by Algorithm 3 with*

$$\beta_k \equiv \beta \geq 6\rho L_{\nabla F} + (2\rho + \frac{2}{3\rho})L_{\nabla f_1}L_{f_2}^2, \quad \rho > 0. \tag{15}$$

*Then, we have $\forall N \geq 1$,*

$$\mathbb{E}\left[\|\mathcal{G}_\mathcal{X}(x^R, \nabla F(x^R), \beta)\|^2\right] \leq \frac{\mathcal{C}_1}{\sqrt{N}}, \quad \mathbb{E}\left[\|f_2(x^R) - u_2^R\|^2\right] \leq \frac{\mathcal{C}_2}{\sqrt{N}}.$$

*(b) Let $T = 1$ and let $\{x^k, z^k\}_{k \geq 0}$ be the sequence generated by Algorithm 4 with $\beta_k \equiv \beta > 0$. Then, we have $\forall N \geq 1$,*

$$\mathbb{E}\left[\|\mathcal{G}_{\mathcal{X}}(x^R, \nabla F(x^R), \beta)\|^2\right] \leq \frac{\mathcal{C}_3}{\sqrt{N}}.$$

*All expectations are taken with respect to all random sequences generated by the respective algorithms and an independent random integer number $R$ uniformly distributed over $\{1, \ldots, N\}$. In both cases, the number of calls to SFO and LMO to get an $\epsilon$-stationary point is upper bounded by $\mathcal{O}(\epsilon^{-2}), \mathcal{O}(\epsilon^{-3})$ respectively.*

**Remark.** *While we can obtain the same complexities without using the linear approximation of the inner function for $T = 2$, it seems necessary to have a parameter-free algorithm as the choice of $\beta$ in (15) depends on the knowledge of the problem parameters. Indeed, the linearization term in (8) helps use to better exploit the Lipschitz smoothness of the gradients get an error bound in the order of $\tau_k^2 \|d^k\|^2$ for estimating the inner function values. Without this term, we are only able to use the Lipschitz continuity of the inner functions and so the error estimate will increase to the order of $\tau_k \|d^k\|$. Hence, we need to choose a larger beta (as in (15)) to reduce $\|d^k\|$ and handle the error term without compromising the complexities. However, this is not the case for $T = 1$ as it can be seen as a two-level problem whose inner function is exactly known (the identity map). In this case, the choice of $\beta$ is independent of the problem parameters with or without the linearization term.*

### 3.2 High-Probability Convergence for $T = 1$

In this subsection, we establish an oracle complexity result with high-probability for the case of $T = 1$. We first provide a notion of $(\epsilon, \delta)$-stationary point and a related tail assumption on the stochastic first-order oracle below.

**Definition 4.** *A point $\bar{x} \in \mathcal{X}$ generated by an algorithm for solving (1) is called an $(\epsilon, \delta)$-stationary point, if we have $\|\mathcal{G}_{\mathcal{X}}(\bar{x}, \nabla F(\bar{x}), \beta)\|^2 \leq \epsilon$ with probability $1 - \delta$.*

**Assumption 4.** *Let $\Delta^{k+1} = \nabla F(x^k) - J_1^{k+1}$ for $k \geq 0$. For each $k$, given $\mathscr{F}_k$ we have $\mathbb{E}[\Delta^{k+1}|\mathscr{F}_k] = 0$ and $\|\Delta^{k+1}\| \big| \mathscr{F}_k$ is $K$-sub-Gaussian.*

The above assumption is commonly used in the literature; see [23, 21, 30, 55]. We also refer to [45] and Appendix E for additional details. The high-probability bound for solving non-convex constrained problems by Algorithm 4 is given below.

**Theorem 5.** *Let Assumptions 1, 2, 4 be satisfied by the optimization problem (1) with $T = 1$. Let $\tau_0 = 1, t_0 = 0, \tau_k = \frac{1}{\sqrt{N}}, t_k = \lceil \sqrt{k} \rceil, \forall k \geq 1$, where $N$ is the total number of iterations. Let $T = 1$ and let $\{x^k, z^k\}_{k \geq 0}$ be the sequence generated by Algorithm 4 with $\beta_k \equiv \beta > 0$. Then, we have $\forall N \geq 1, \delta > 0$, with probability at least $1 - \delta$,*

$$\min_{k=1,\ldots,N} \left\|\mathcal{G}_{\mathcal{X}}(x^k, \nabla F(x^k), \beta)\right\|^2 \leq \mathcal{O}\left(\frac{K^2 \log(1/\delta)}{\sqrt{N}}\right)$$

*Therefore, the number of calls to SFO and LMO to get an $(\epsilon, \delta)$-stationary point is upper bounded by $\mathcal{O}(\epsilon^{-2} \log^2(1/\delta)), \mathcal{O}(\epsilon^{-3} \log^3(1/\delta))$ respectively.*

**Remark.** *To the best of our knowledge, the above result is (i) the first high-probability bound for one-sample stochastic conditional gradient-type algorithm for the case of $T = 1$, and (ii) the first high-probability bound for constrained stochastic optimization algorithms in the non-convex setting; see Appendix J of [32].*

## 4 Proof Sketch of Main Results

In this section, we only present the proof sketch. The complete proofs are provided in the appendix. For convenience, let $u_{T+1} = x$, and we denote $H_k$ as the function value of the subproblem at step $k$, $y^k$ as the optimal solution of the subproblem i.e.,

$$H_k(y) := H(y; x^k, z^k, \beta_k), \quad y^k = \arg\min_{y \in \mathcal{X}} H_k(y). \tag{16}$$

Then, the proof of Theorem 2 proceeds via the following steps:

1. We first leverage the merit function $W_k := W_{\alpha,\gamma}(x^k, z^k, u^k)$ defined in (9) with appropriate choices of $\alpha, \gamma$ for any $\beta > 0$ to obtain

$$W_{k+1} - W_k \leq -\frac{\tau_k}{2}\left(\beta\left[\|d^k\|^2 + \sum_{i=1}^{T}\|f_i(u_{i+1}^k) - u_i^k\|^2\right] + \frac{\beta}{20L_{\nabla F}^2}\|\nabla F(x^k) - z^k\|^2\right)$$
$$+ \mathbf{R}_k + \tau_k\left(\frac{12}{5} + \frac{20L_{\nabla F}^2}{\beta^2}\right)\left(H_k(\tilde{y}^k) - H_k(y^k)\right), \quad \forall k \geq 0$$

   where $\mathbf{R}_k$ is the residual term (see (31)) and $\mathbb{E}[\mathbf{R}_k|\mathscr{F}_k] \leq \hat{\sigma}^2\tau_k^2$, as shown in Proposition 3.

2. Telescoping the above inequality, in Lemma 11 we obtain the following:

$$\sum_{k=1}^{N}\tau_k\left[\beta\left(\|d^k\|^2 + \sum_{i=1}^{T}\|f_i(u_{i+1}^k) - u_i^k\|^2\right) + \frac{\beta}{20L_{\nabla F}^2}\|\nabla F(x^k) - z^k\|^2\right]$$
$$\leq 2W_0 + 2\sum_{k=0}^{N}\mathbf{R}_k + \left(\frac{24}{5} + \frac{40L_{\nabla F}^2}{\beta^2}\right)\sum_{k=0}^{N}\tau_k\left(H_k(\tilde{y}^k) - H_k(y^k)\right), \quad \forall N \geq 1.$$

3. To further control the error term $H_k(\tilde{y}^k) - H_k(y^k)$ introduced by the ICG method, we set $t_k$, the number of iterations in ICG method at step $k$, to $\lceil\sqrt{k}\rceil$. By Lemma 8, we therefore have

$$H_k(\tilde{y}^k) - H_k(y^k) \leq \frac{2\beta D_{\mathcal{X}}^2(1+\delta)}{t_k + 2} \leq \frac{2\beta D_{\mathcal{X}}^2(1+\delta)}{\sqrt{k}}, \quad \forall k \geq 1.$$

   Also, with the choice of $\tau_k = \frac{1}{\sqrt{N}}$ and $z^0 = 0$, we can conclude that

$$\sum_{k=0}^{N}\tau_k\left(H_k(\tilde{y}^k) - H_k(y^k)\right) \leq \frac{2\beta D_{\mathcal{X}}^2(1+\delta)}{\sqrt{N}}\sum_{k=1}^{N}\frac{1}{\sqrt{k}} \leq 4\beta D_{\mathcal{X}}^2(1+\delta).$$

4. Then, taking expectation of both sides and by the definition of random integer $R$, we have

$$\mathbb{E}\left[\beta\left(\|d^R\|^2 + \sum_{i=1}^{T}\|f_i(u_{i+1}^R) - u_i^R\|^2\right) + \frac{\beta}{20L_{\nabla F}^2}\|\nabla F(x^R) - z^R\|^2\right] \leq 2W_0 + \mathcal{B},$$

   $\forall N \geq 1$, where $\mathcal{B}$ is a constant depending on the problem parameters $(\beta, \sigma^2, L, D_{\mathcal{X}}, T, \delta)$.

5. As a result, we can obtain (13) and (14) by noting that $\forall k \geq 1$

$$\|\mathcal{G}(x^k, \nabla F(x^k), \beta)\|^2 \leq 2\beta^2\|d^k\|^2 + 2\beta^2\left\|\Pi_{\mathcal{X}}\left(x^k - \frac{1}{\beta}\nabla F(x^k)\right) - \Pi_{\mathcal{X}}\left(x^k - \frac{1}{\beta}z^k\right)\right\|^2$$
$$\leq 2\beta^2\|d^k\|^2 + 2\|\nabla F(x^k) - z^k\|^2.$$

   where the second inequality follows the non-expansiveness of the projection operator.

The proofs of Theorems 3 and 5 follow the same argument with appropriate modifications. The high-probability convergence proof of Theorem 5 mainly consists of controlling the tail probability of the residual term $\mathbf{R}_k$ being large.

## 5  Discussion

In this work, we propose and analyze projection-free conditional gradient-type algorithms for constrained stochastic multi-level composition optimization of the form in (1). We show that the oracle complexity of the proposed algorithms is level-independent in terms of the target accuracy. Furthermore, our algorithm does not require any increasing order of mini-batches under standard unbiasedness and bounded second-moment assumptions on the stochastic first-order oracle, and is parameter-free. Some open questions for future research: (i) Considering the one-sample setting, either improving the LMO complexity from $\mathcal{O}(\epsilon^{-3})$ to $\mathcal{O}(\epsilon^{-2})$ for general closed convex constraint sets or establishing lower bounds showing that $\mathcal{O}(\epsilon^{-3})$ is necessary while keeping the SFO in the order of $\mathcal{O}(\epsilon^{-2})$, is extremely interesting; and (ii) Providing high-probability bounds for stochastic multi-level composition problems ($T > 1$) and under sub-Gaussian or heavy-tail assumptions (as in [32, 31]) is interesting to explore.

## Acknowledgment

The authors are grateful to anonymous reviewers for their constructive comments that greatly improved the presentation of this paper. TX was partially supported by a seed grant from the Center for Data Science and Artificial Intelligence Research, UC Davis and National Science Foundation (NSF) grant CCF-1934568. KB was partially supported by a seed grant from the Center for Data Science and Artificial Intelligence Research, UC Davis and NSF grant DMS-2053918. SG was partially supported by an NSERC Discovery Grant.

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
