# OpenReview forum: "A Projection-free Algorithm for Constrained Stochastic Multi-level Composition Optimization"
_NeurIPS.cc/2022/Conference — NeurIPS 2022 Accept_

### Official Review · Reviewer_R3Vn · 2022-06-29

**Rating:** 5
**Confidence:** 4
**Soundness:** 4 excellent
**Presentation:** 4 excellent
**Contribution:** 2 fair

**Summary:**

In this work, the authors proposed a projection-free algorithm (LiNASA-ICG) to solve the constrained stochastic multi-level optimization problem. The oracle complexities are derived for the algorithm and the algorithm is shown to be parameter-free. In addition, improvements in the T=1,2 case are provided and a high-probability bound is derived for the T=1 case.

**Questions:**

Major:
(1) The comparison to [4] is not sufficient. Maybe I missed something, I think the major difference between the proposed algorithm and the algorithm in [4] is the generation of the next iteration point. Namely, sub-problem (5) in this work and sub-problem (3) in [4], which is the same as sub-problem (2) in this work. Since both this work and [4] aimed at solving (2), it is unclear how much advantage one can gain by utilizing the "projection-free" ICG procedure. Therefore, I would suggest the authors provide a more detailed comparison to [4] and numerical experiments, which will also help highlight the contributions of this work.


Minor:
(1) It would be better to clarify the difference between "stochastic conditional gradient-type algorithms" and "stochastic gradient-type algorithms". I guess the second term refers to the gradient descent-type algorithms?

(2) The sentence at ln.135-138 is confusing and hard to parse. I would suggest the authors revise this sentence.

(3) Ln.144 and ln.146: "assume" -> "assumed".

(4) Ln.187-188: I do not understand why \xi_i could be independent of u_i? I think they are related by relation (8) and cannot be independent. It would be better if more details can be included.

(5) Ln.251: "beta" -> "\beta".

(6) It would be better if the authors could mention the difficulty of deriving high-probability bounds for the general T case. Alternatively, I wonder if it is possible to derive a (possibly worse) high-probability bound using the proof technique of the T=1 case.

**Limitations:**

See my comments in the "questions" section.

**Strengths And Weaknesses:**

The topic of this paper is interesting to audiences in the optimization and machine learning fields. As the authors mentioned, the proposed algorithm has the advantage that it is projection-free and parameter-free. In addition, the algorithm does not require mini-batch, which will enable its application to the online optimization case. The paper is also well written and easily to follow.

My major concern is the contributions compared to [4]. It seems to me that many components in this work were already proposed in [4]. For example, the LiNASA algorithm and the construction of the merit function. Additionally, the comparison to [4] is not sufficient, which makes it hard to judge the contribution of this work. Hence, I would like to stay borderline and wait for the authors' response at this time.

---

> ### Author Response · Authors · 2022-08-02
> **Answers to Reviewer R3Vn**
>
> We thank the reviewer for the useful comments.  We answer the questions below.
>
> **Comparison to [4]**:   At a methodological level, while incorporating conditional gradient methods to nested averaging-based multi-level composition optimization seems rather immediate, the theoretical analysis is novel, and the resulting consequences are far-reaching. In particular, we have the following advantages:
> - Our approach and analysis help us to avoid having to use increasing order mini-batches under much relaxed conditions than those existing in the literature (even for $T=1$).
> - Even for the case of $T=1$, we provide high-probability bounds for projection-free methods which have been lacking in the literature.
> - Our algorithm is a single-time scale algorithm and does not require knowledge of problem parameters for implementation.
>
> Furthermore, our analysis is different and requires more effort in 2 aspects:
> - Our proof leverages a different merit function compared to [4] (highlighted in the revision in lines 194-196) . Adding $\Vert \nabla F(x) - z \Vert^2$ to the merit function is essential in our proof due to the ICG routine.
> - We emphasize that unlike the ICG routine in previous works [3, 39] which requires terminating criteria in terms of FW-gap, the number of iterations, $t_k$, required to run ICG is given by $t_k = \lceil\sqrt{k}\rceil$. That is, we require more precise solutions for the ICG routine, only for later outer iterations.
>
> We sincerely hope that the above mentioned important finer details highlight the novelty of our work, in which case, we would greatly appreciate if the scores could be increased to reflect it. Please also feel free to reach out to us in the discussion phase should you have additional questions.
>
> **Experiments**: We have added preliminary experiments in the Appendix on the matrix-valued single-index model under low-rank constraints, showing the efficiency of our method when T=1 compared to 1-SFW in the online setting without increasing batch sizes.
>
> **Minor**
>
> 1. Done as per reviewer request.
> 2. Done as per reviewer request.
> 3. Done as per reviewer request.
> 4. We refer the reviewer to the (non)-oblivious setting introduced in [54]. We will add more explanation in the camera-ready version where we will have an additional page to address the reviews.
> 5. Done as per reviewer request.
> 6. The difficulty of deriving high probability bounds for the general T-level problems arises from the multiplicative noise at each level. The deeper the level of the stochastic composition problems, the heavier the tails of the moving averaging multiplicative gradient estimate of the objective $F$. Obtaining concentration results in this case is the main challenge to overcome to get high-probability results.
>
> **Reference**
>
> [3] Balasubramanian, et al. "Zeroth-order nonconvex stochastic optimization: Handling constraints, high dimensionality, and saddle points"
>
> [4] Balasubramanian, et al. "Stochastic multilevel composition optimization algorithms with level-independent convergence rates"
>
> [39] Qu, et al. "Non-convex conditional gradient sliding"
>
> [54] Zhang, et al.. "One-sample Stochastic Frank-Wolfe"

---

> > ### Author Response · Authors · 2022-08-06
> > **Looking forward to your feedback**
> >
> > Dear Reviewer R3Vn,
> >
> > As you had mentioned "I would like to stay borderline and wait for the authors' response at this time" in your initial review, and
> > as we are half way through the discussion period already, we were wondering if you have any further questions based on our initial author response. Please let us know if you have additional questions and we will be happy to answer them.
> >
> > sincerely,
> >
> > Authors.

---

> > ### Comment · Reviewer_R3Vn · 2022-08-07
> > **Post-rebuttal**
> >
> > I would like to thank the authors for responding to my quesitons! I have read all reviews and responses, and I would be happy to increase my score by 1.

---

### Official Review · Reviewer_41Ug · 2022-07-06

**Rating:** 5
**Confidence:** 3
**Soundness:** 3 good
**Presentation:** 3 good
**Contribution:** 2 fair

**Summary:**

This paper studies stochastic multi-level composition optimization problem, and especially focus on the class of projection-free algorithms. By combining the linearized NASA in existing works on multi-level composition problem and the existing conditional gradient algorithms, this paper propose a new algorithm that theoretically outperform other existing projection-free algorithms in the stochastic multi-level composition optimization problem.

**Questions:**

1. Could the author provide some insights for their proofs on why the algorithms can work without increasing batchsize, comparing to existing results.?

2. I am confused on why the authors says that the bound in equation (13) is independent of T. From equation (42), it seems like the the bound should be exponentially dependent on T (There are summations over T and comulative product over T in the constant \hat \sigma^2.)?

**Ethics Review Area:**

["I don’t know"]

**Limitations:**

See my comments above.

**Strengths And Weaknesses:**

Strength:
1. I think one major theoretical novelty in this paper is that their proposed algorithm does not need an increasing batchsize and extra assumptions.

Weakness:
1. This paper appears to be incremental, and the problem setting seems contrive to me. First, it seems that the independence of the levels is due to the linearized NASA technique in existing works. Second, the theoretical analysis (Using gradient mapping as criterion and the moving averaging technique) seems standard in the existing literatures on SCG. Therefore, it looks like this paper is a direct combination of the existing results in multi-level composition optimization and conditional gradient algorithms.

---

> ### Author Response · Authors · 2022-08-02
> **Answers to Reviewer 41Ug**
>
> We thank the reviewer for the useful comments. We answer the questions below.
>
> **... paper appears to be incremental...**: At a methodological level, while incorporating conditional gradient methods to nested averaging based multi-level composition optimization seems rather immediate, the theoretical analysis is novel, and the resulting consequences are far-reaching. In particular, we have the following advantages:
> - Our approach and analysis help us to avoid having to use increasing order mini-batches.
> - Even for the case of T=1, we provide high-probability bounds for projection-free methods which have been lacking in the literature.
>
> Furthermore, our analysis is different and requires more effort in 2 aspects:
> - Our proof leverages a different merit function compared to [4] (highlighted in the revision). Adding $\Vert \nabla F(x) - z \Vert^2$ to the merit function is essential in our proof due to the ICG routine.
> - We emphasize that unlike the ICG routine in previous works [3, 39] which requires terminating criteria in terms of FW-gap, the number of iterations, $t_k$, required to run ICG is given by $t_k = \lceil\sqrt{k}\rceil$. That is, we require more precise solutions for the ICG routine, only for later outer iterations.
> - Furthermore, existing 1-sample SFW algorithms using moving-average techniques are two-time scale algorithms, whereas our approach is a single-time scale algorithm.
>
> We request the reviewer to take the above mentioned finer details into consideration while evaluating the novelty.
>
> **Questions**:
> 1. The moving averaging technique helps to avoid increasing batchsize. One important issue in stochastic optimization is to control the variance associated with the gradient estimators. When $T=1$ and the problem is convex or unconstrained nonconvex, we can control the summation of the error terms by only assuming the boundedness of the variance/second moment of gradient estimators. However, when the problem in constrained and nonconvex, we have an additional error term when updating the main iterate and thus, we need to reduce the variance of gradient estimators. Taking mini-batch of samples is a variance-reduction technique that has its own limitations such as not being appropriate for online learning. Using the history of generated stochastic gradients and forming a convex combination of them as gradient estimators is another technique to reduce the variance. Furthermore, having the multi-level structure imposes another source of error in estimating the gradient. By using the weighted average of stochastic gradients, we are able to reduce the variance of gradient estimators in this multi-level setting. As shown in the paper, this variance goes to zero when the number of iterations increases.
>
> 2. Please note that the dependency of the target-accuracy is independent of $T$. This is what we refer to as level-independent accuracy.  We use $\mathcal{O}_T$ to represent the fact that the involved constant might depend on $T$ (which could be exponential in the worst case, but for specific problems could be controlled more explicitly).
>
> To provide some context, in the unconstrained or projection-based setting, the work of [50] provides algorithms for which the $\epsilon$ depends exponentially on $T$. The work of [4, 7, 19, 43] removed this exponential dependence under various assumptions and developed algorithms with level-independent rates of convergence (with constants depending on $T$ in a similar manner as our result).
>
> **Reference**
>
> [3] Balasubramanian, et al. "Zeroth-order nonconvex stochastic optimization: Handling constraints, high dimensionality, and saddle points"
>
> [4] Balasubramanian, et al. "Stochastic multilevel composition optimization algorithms with level-independent convergence rates"
>
> [7] Chen, et al. "Solving stochastic compositional optimization is nearly as easy as solving stochastic optimization"
>
> [19] Ghadimi, et al. "A single timescale stochastic approximation method for nested stochastic optimization"
>
> [39] Qu, et al. "Non-convex conditional gradient sliding"
>
> [43] Ruszczynski. "A stochastic subgradient method for nonsmooth nonconvex multilevel composition optimization"
>
> [50] Yang, et al. "Multilevel stochastic gradient methods for nested composition optimization".

---

> > ### Comment · Reviewer_41Ug · 2022-08-07
> > **Thanks for your reply!**
> >
> > I have read the authors' reply and would like to stay with my original score. Though I am still concerned with the claim that (13) is level-independent, it seems common in related works.
> >
> > For the novelty, like I said, I did not see any new idea from algorithmic aspect. The authors claim that their main novelty is in the theoretical analysis: 1. Thery can work with small mini-batchsize and 2. provide high-probablity bound. The first point seems to be important though the main idea is mainly attributed to the well-known moving averaging technique.  The second one seems rather direct and less important from my point of view.  But of course, I am not familar with related works and hence not able to give a confident judgement on these points.

---

> > > ### Author Response · Authors · 2022-08-07
> > > **Clarification**
> > >
> > > 1) While the main idea is based on moving-average, the few other papers that considered moving-average estimators in the context of conditional gradient algorithms (i.e., [1], [34], [54]) made several restrictive assumptions even for the case of $T=1$ in their analysis. Please see lines 112-124 for an elaborate discussion. Our analysis, on the other hand, is based on standard assumptions in the stochastic optimization literature and  holds for $T\geq 1$ as well!
> > >
> > > 2) **The second one (high-probability bound) seems rather direct and less important from my point of view**: We respectfully disagree with this comment! A majority of the works in stochastic optimization in the last decade (and in particular on stochastic conditional gradient algorithms) only provide expectation bounds. High-probability bounds in stochastic optimization are **rare but provide much stronger guarantees** considering the problem is actually stochastic!
> > > See for example the abstract of [1] which mentions that **More nuanced high probability guarantees are rare**. Furthermore, several recent works have started to explore high-probability bounds for non-convex stochastic optimization in the past two years: [2]-[5] and references therein, for an incomplete list of recent works.  We emphasize that all the above works do not consider stochastic conditional gradient algorithms which is our main focus.
> > >
> > >
> > > [1] Davis, Damek, Dmitriy Drusvyatskiy, Lin Xiao, and Junyu Zhang. "From low probability to high confidence in stochastic convex optimization." Journal of machine learning research 22, no. 49 (2021).
> > >
> > > [2] Kavis, Ali, Kfir Levy, and Volkan Cevher. "High Probability Bounds for a Class of Nonconvex Algorithms with AdaGrad Stepsize." 10th International Conference on Learning Representations (ICLR), 2022.
> > >
> > > [3] Ward, R., Wu, X., & Bottou, L. (2020). Adagrad stepsizes: Sharp convergence over nonconvex landscapes. The Journal of Machine Learning Research, 21(1), 9047-9076.
> > >
> > > [4] Lei, Yunwen, and Ke Tang. "Learning rates for stochastic gradient descent with nonconvex objectives." IEEE Transactions on Pattern Analysis and Machine Intelligence 43, no. 12 (2021): 4505-4511.
> > >
> > > [5] Li, S., & Liu, Y. (2022, June). High Probability Guarantees for Nonconvex Stochastic Gradient Descent with Heavy Tails. In International Conference on Machine Learning (pp. 12931-12963). PMLR.

---

### Official Review · Reviewer_nJmK · 2022-07-10

**Rating:** 5
**Confidence:** 4
**Soundness:** 3 good
**Presentation:** 3 good
**Contribution:** 2 fair

**Summary:**

The authors propose a projection-free method to solve the multi-level optimization problem, and analyse the number of calls to the stochasic first-order oracle and the linear-minimization oracle. The complexity is comparable with exisiting results for single-level problems. For single-level case (T=1), they also give a high-probability convergence result. Another merit of the proposed method is that it dose not require large batchsize and is parameter-free.

**Questions:**

1. Since this paper is about multi-level optimization, why not compare the complexity of the algorithm with other multi-level methods, such as [4],[7],[50],[52]?

**Limitations:**

No problems here. The authors discuss the limitations in Section 5, and I think they are proper.

**Strengths And Weaknesses:**

Strengths:

1. The proposed method is projection-free and parameter-free. It also does not require any increasing order of mini-batches. This paper is clear-written and analyses are supported by proofs that are sound to me.

Weaknesses:

1. As pointed out by the authors, the LMO complexity of the proposed method is worse than the single-level problem, i.e., $\mathcal{O}(\epsilon^{-3})$ vs $\mathcal{O}(\epsilon^{-2})$. Whether this rate is improvable is an important problem.

2. The algorithm proposed is not novel enough. As mentioned in Section 2, the techniques used are already known in other literature. Especially, the method is a bit similar to existing multi-level work, i.e., [4], although the latter uses the projection operation to ensure the final point in the domain.

3. Experiment results are not provided to show the practical performance of the proposed method.

---

> ### Author Response · Authors · 2022-08-02
> **Answers to Reviewer nJmK**
>
> We thank the reviewer for the useful comments.  We answer the questions below.
>
> **LMO complexity**:  It would be very interesting to either improve the LMO complexity or establish the lower bound of LMO complexity in the general one-sample regime, under the current set of assumptions. Obtaining such lower bounds is beyond the scope of this work.
>
> However, the current LMO complexity could be immediately improved under further assumptions on the constraint set $\mathcal{X}$. In particular, note that the ICG routine is **deterministic** in nature. Hence, further advances in the analysis of deterministic conditional gradient methods under additional assumptions on the constraint set $\mathcal{X}$, for example[16,18], would immediately help to improve the overall LMO complexity.
>
> **Novelty**:
>
> - While leveraging conditional gradient sliding methods to replace the projection operator is standard by now in the literature, our work is the first one to incorporate this method into multi-level problems and establish theoretical results.
> - Even for the case of T=1, our algorithm and analysis has several advantages: (1) it avoids having to use increasing mini-batches under much milder assumptions that those established in the literature before, (2) it helps to obtain high-probability bounds which were lacking in the literature for stochastic conditional gradient type algorithms, in particular for algorithms that do not use increasing mini-batches.
> - In addition, we emphasize that unlike the ICG routine in previous works [3, 39] which requires terminating criteria in terms of FW-gap, the number of iterations, $t_k$, required to run ICG is given by $t_k = \lceil\sqrt{k}\rceil$. That is, we require more precise solutions for the ICG routine, only for later outer iterations.
> - Moreover, our proof leverages a different merit function compared to [4] (highlighted in the revision in lines 194-196) which essentially distinguishes our contributions from [4].
>
> **Experiments**: Taking the reviewer's comment into consideration, we have now added preliminary experiments in the Appendix on the matrix-valued single-index model under low-rank constraints, showing the efficiency of our method when T=1 compared to 1-SFW in the online setting without increasing batch sizes. However, we also emphasize that our main contribution in this work is theoretical in nature.
>
> **Questions**:  The focus of our work is on projection-free stochastic condition gradient-type algorithms. As is common in the literature on such algorithms, we report and compare SFO and LMO complexities. The references [4], [7], [50] and [52] mentioned, either consider unconstrained or projection-based algorithms. Hence, we have discussed these papers elaborately in lines 126-153.
>
> **Reference**
>
> [3] Balasubramanian, et al. "Zeroth-order nonconvex stochastic optimization: Handling constraints, high dimensionality, and saddle points"
>
> [4] Balasubramanian, et al. "Stochastic multilevel composition optimization algorithms with level-independent convergence rates"
>
> [7] Chen, et al. "Solving stochastic compositional optimization is nearly as easy as solving stochastic optimization"
>
> [16] Dan, et al. "Faster rates for the frank-wolfe method over strongly-convex sets"
>
> [18] Dan, et al. "Frank-Wolfe with a nearest extreme point oracle"
>
> [39] Qu, et al. "Non-convex conditional gradient sliding"
>
> [43] Ruszczynski. "A stochastic subgradient method for nonsmooth nonconvex multilevel composition optimization"
>
> [50] Yang, et al. "Multilevel stochastic gradient methods for nested composition optimization".
>
> [52] Zhang, et al. "Multilevel composite stochastic optimization via nested variance reduction."

---

> > ### Comment · Reviewer_nJmK · 2022-08-07
> > **Thanks for the response!**
> >
> > Thank you for the detailed explanation.
> >
> > The response address some of my concerns and I decide to keep my original score.

---

> > > ### Author Response · Authors · 2022-08-07
> > > **Thank you**
> > >
> > > Thanks for the acknowledgement of the response. If you could let us know which concerns still remain unaddressed or have additional questions, we would be happy to clarify them!

---

### Meta-Review · Area_Chair_WLXp · 2022-08-21

**Recommendation:** Accept
**Confidence:** Less certain

**Metareview:**

There is general agreement that this paper is a borderline accept and after my own reading I feel similar.

**Award:**

No

---

### Decision · Program_Chairs · 2022-09-14

Accept